# Electrostatic force promoted intermolecular stacking of polymer donors toward 19.4% efficiency binary organic solar cells

Zirui Gan[1], Liang Wang[1], Jinlong Cai[1], Chuanhang Guo[1], Chen Chen[1], Donghui Li[1], Yiwei Fu[1], Bojun Zhou[1], Yuandong Sun[1], Chenhao Liu[1], Jing Zhou[1], Dan Liu[1], Wei Li[1] & Tao Wang [1,2] ✉

Conjugated polymers are generally featured with low structural order due to their aromatic and irregular structural units, which limits their light absorption and charge mobility in organic solar cells. In this work, we report a conjugated molecule INMB-F that can act as a molecular bridge via electrostatic force to enhance the intermolecular stacking of BDT-based polymer donors toward efficient and stable organic solar cells. Molecular dynamics simulations and synchrotron X-ray measurements reveal that the electronegative INMB-F adsorb on the electropositive main chain of polymer donors to increase the donor-donor interactions, leading to enhanced structural order with shortened π-π stacking distance and consequently enhanced charge transport ability. Casting the non-fullerene acceptor layer on top of the INMB-F modified donor layer to fabricate solar cells via layer-by-layer deposition evidences significant power conversion efficiency boosts in a range of photovoltaic systems. A power conversion efficiency of 19.4% (certified 18.96%) is realized in PM6/L8-BO binary devices, which is one of the highest reported efficiencies of this material system. The enhanced structural order of polymer donors by INMB-F also leads to a six-fold enhancement of the operational stability of PM6/L8-BO organic solar cells.

Organic solar cells (OSCs) have attracted enormous attention due to their advantages including light-weight, flexibility, solution processability and semitransparency[1-3]. In recent years, the power conversion efficiency (PCE) of OSCs has climbed significantly with the advent of novel organic semiconductors as well as efforts on device engineering, with PCEs exceeding 19% in single-junction OSCs and over 20% in tandem OSCs[4-11]. Nevertheless, attributing to the weak non-covalent bond interactions among photovoltaic materials, OSCs are generally featured with photoactive layers having a low structural order and thus exhibit inferior charge generation and collection efficiencies compared to their inorganic counterparts (i.e. Si or perovskite solar cells)[12-15].

Regards the above critical issues in OSCs, understanding the molecular packings and constructing ideal nanoscale morphology in OSCs has been believed to be the most effective method to optimize the power conversion process toward higher PCE[16-19]. As such, throughout the history of morphology optimization of OSCs, various approaches such as solvent solution design, thermal annealing and solvent annealing have been developed, which direct to multiple morphological metrics modulation wisdoms, including acquiring moderate phase separation and domain size among donors and acceptors to allow efficient exciton dissociation and charge transport, converting molecular orientation from edge-on to face-on or building

[1]School of Materials Science and Engineering, Wuhan University of Technology, Wuhan, China. [2]School of Materials and Microelectronics, Wuhan University of Technology, Wuhan, China. ✉e-mail: twang@whut.edu.cn

favorable vertical components distributions to enable efficient charge transport toward electrodes[20–25].

As a facile approach to tune morphology, utilizing additive molecules to adjust the molecular interactions among photovoltaic components has been demonstrated to be an efficient method to optimize the exciton dissociation, charge transport and collection processes[26–38]. For example, Hou et al. designed a series of volatilizable molecules that were able to interact with non-fullerene acceptors (NFAs) and promote molecular packing upon their volatilization after thermal annealing (TA)[26]. Non-volatilizable solid additives, e.g. BF7, have also been explored to optimize the nano-crystallites of Y6 to improve morphology of the active layer and therefore improve device performance and thermal stability[27]. However, current works only demonstrated the beneficial effects of these additives on the molecular order of small molecular NFAs[30–36], their effects on polymer donors are less investigated and only limited works revealed that they can tune the domain size and phase distribution of polymer donors[37,39], but not the molecular packing. Different to the rigid small molecular NFAs, conjugated polymer donors generally have lower structural order with larger π-π stacking distance that is inferior for charge transport[40], and hence exploring approaches to improve the ordering of polymer donors is very promising at further boosting the performance of OSCs, yet is extremely challenging.

In this work, we designed and synthesized conjugated molecule 2-(4-fluoro-2-methyl benzylidene)−1H-indene-1,3(2H)-dione (INMB-F), which can act as a molecular bridge to modulate the intermolecular interactions among a range of BDT-based polymer donors, including PTB7-Th, D18-Cl and PM6, and enhances their intermolecular stacking in solid thin films. Molecular dynamics simulations and synchrotron X-ray scattering revealed that the electronegative INMB-F could interact with the electropositive main chains of polymer donors via electrostatic force (dipole-dipole interaction), and thus reduce the intermolecular interaction energy to allow stronger donor-donor connections, resulting in enhanced π-π stacking as well as shorter stacking distance. As the results, four binary photovoltaic systems, PM6/BTP-4F-C5-16 (C5-16), PM6/L8-BO, D18-Cl/L8-BO and PTB7-Th/C5-16, all obtained enhanced fill factors (FFs) and short-circuit current density ($J_{SC}$), due to enhanced light absorption and charge transport. A PCE of 19.4% (certified 18.96%) was realized in PM6/L8-BO binary OSCs, which is among the highest reported efficiencies of this material system. Moreover, the enhanced donor-donor interactions by INMB-F also leads to a six-fold enhancement of the operational stability of PM6/L8-BO OSCs in air.

## Results

### Materials and optoelectronic characterization

The chemical structures and molecular weights of polymer donors, synthesis routes and characterization details of the conjugated molecule INMB-F are shown in Fig. 1a and Supplementary Fig. 1-5. INMB-F contains a 1H-indene-1,3(2H)-dione moiety connecting a benzaldehyde unit with methyl and fluorine attached on. This structure feature allows INMB-F good solubility in common solvents including dichloromethane (DCB), chloroform (CF) and chlorobenzene (CB), together with a relatively high volatility of around 120 °C for 10 min (Supplementary Fig. 6). The two-dimensional (2D) grazing-incidence wide-angle X-ray scattering (GIWAXS) patterns of INMB-F film casting from CF is shown in Supplementary Fig. 7, where multiple isotropic diffraction peaks with strong intensity can be found, suggesting that INMB-F possesses a remarkable crystallinity in solid film state. INMB-F has good miscibility with either polymer donors or NFAs, featuring a small Flory-Huggins interaction parameter ($\chi$) (Supplementary Fig. 8 and Supplementary Table 1). The electrostatic potential surfaces (ESP) of INMB-F and a range of state-of-the-art polymer donors including PM6[41], D18-Cl[42] and PTB7-Th[43] were calculated by density functional theory (DFT). According to the calculation results shown in Fig. 1b, the conjugated main chains of these polymer donors show strong electronegativity while the conjugated units of INMB-F show strong electropositivity, denoting that INMB-F can potentially interact with the conjugated main chains of the above polymer donors and influence their molecular organization during the film-formation process. This can be further validated by the Raman spectroscopy measurement (Supplementary Fig. 9), where the neat PM6 film shows negligible shifts at 1423 cm⁻¹ (attributing to the flexural vibration of C-H in the alkyl chain) but red-shift from 1531 cm⁻¹ to 1536 cm⁻¹ for the C = C stretching vibration of coupled BDT and BDD units, validating that INMB-F has interacted with polymers along their conjugated main chains, consistent with our ESP results[44]. We also investigated the influence of INMB-F with other conjugated polymer donors PTQ10 and P3HT that don't contain the BDT and BDD units. As expected, negligible difference was found in the absorption spectra of PTQ10 and P3HT upon the addition of INMB-F (Supplementary Fig. 10), further confirming that INMB-F interacts with these BDT-based polymers through their BDT and BDD units.

### INMB-F directed optical and morphological properties of polymer donors

To explore the effect of INMB-F on structural order of the BDT-based polymer donors, 10 wt.% INMB-F was first introduced into PM6, PTB7-Th and D18-Cl CF solutions to cast films, following with a routinely used 80 °C thermal annealing (TA) to allow full evaporation of CF. From their UV–vis absorption spectra shown in Fig. 1c−e, it is apparent that the addition of INMB-F enables enhanced (0-0)/(0-1) intensity ratios with obvious red-shift of their (0-0) absorption peaks, together with stronger aggregation in their atomic force microscope (AFM) images (as shown in Supplementary Fig. 11), indicating that INMB-F has modulated the aggregation state of these polymer donors[45]. We also found further increasing the INMB-F content to 20 wt.% could lead to a significantly reduced absorption intensity (Supplementary Fig. 12), suggesting the excessive addition of INMB-F could hamper the organization of polymer donors and generate severe self-aggregation, which can be confirmed by the 2D GIWAXS patterns (Supplementary Fig. 13, where isotropic diffraction peaks of INMB-F appear) and AFM images (Supplementary Fig. 14, within which the white spots are self-aggregated INMB-F).

Without the presence of INMB-F, neat PM6 film presents a crescent-shaped π-π stacking diffraction peak at $q_z = 1.68$ Å⁻¹ in the out-of-plane (OOP) direction (Fig. 2a), two lamellar diffraction peaks at $q_z = 0.28$ Å⁻¹ in OOP and $q_{xy} = 0.31$ Å⁻¹ in the in-plane (IP), respectively. When PM6 was processed with the addition of 10% INMB-F, PM6 not only exhibits a significantly enhanced lamellar diffraction intensity in OOP ($q_z = 0.28$ Å⁻¹) with Herman's orientation parameter changed from 0.47 to 0.60 (Supplementary Fig. 15)[46], but also shows a new diffraction peak at $q_z = 0.9$ Å⁻¹ in OOP together with a largely increased q value from 1.68 Å⁻¹ to 1.72 Å⁻¹ that is associated with π-π stacking, suggesting the existence of INMB-F can greatly affect the crystallization of PM6 and shorten the π-π stacking distance (d) from 3.74 to 3.65 Å ($d = 2\pi/q$). Further increasing the INMB-F content to 20% results in reduced diffraction peaks of PM6 but the emergence of a number of diffraction peaks of INMB-F, suggesting the excessive addition and crystallization of INMB-F could suppress the crystallization of PM6. To evaluate how the full volatilization of INMB-F would affect PM6, a sample was annealed at 120 °C to completely remove INMB-F, with GIWAXS in Supplementary Fig. 13c demonstrates reduced crystallinity of PM6, implying that INMB-F needs to exist in the film to act as the molecular bridge to modulate the intermolecular interaction of PM6 molecules.

The 2D GIWAXS patterns and 1D profiles of D18-Cl and PTB7-Th films with or without 10% INMB-F are shown in Fig. 2c−f. Same as PM6, with the addition of INMB-F, these donors all show stronger and sharper (010) π-π stacking peak in the OOP direction, alongside with significantly increased q values of the π-π stacking peak from 1.55 Å⁻¹ to 1.70 Å⁻¹ in the PTB7-Th film and from 1.64 Å⁻¹ to 1.70 Å⁻¹ in the D18-Cl film, translating to greatly significantly reduced π-π stacking distance from 4.05 to 3.70 Å,

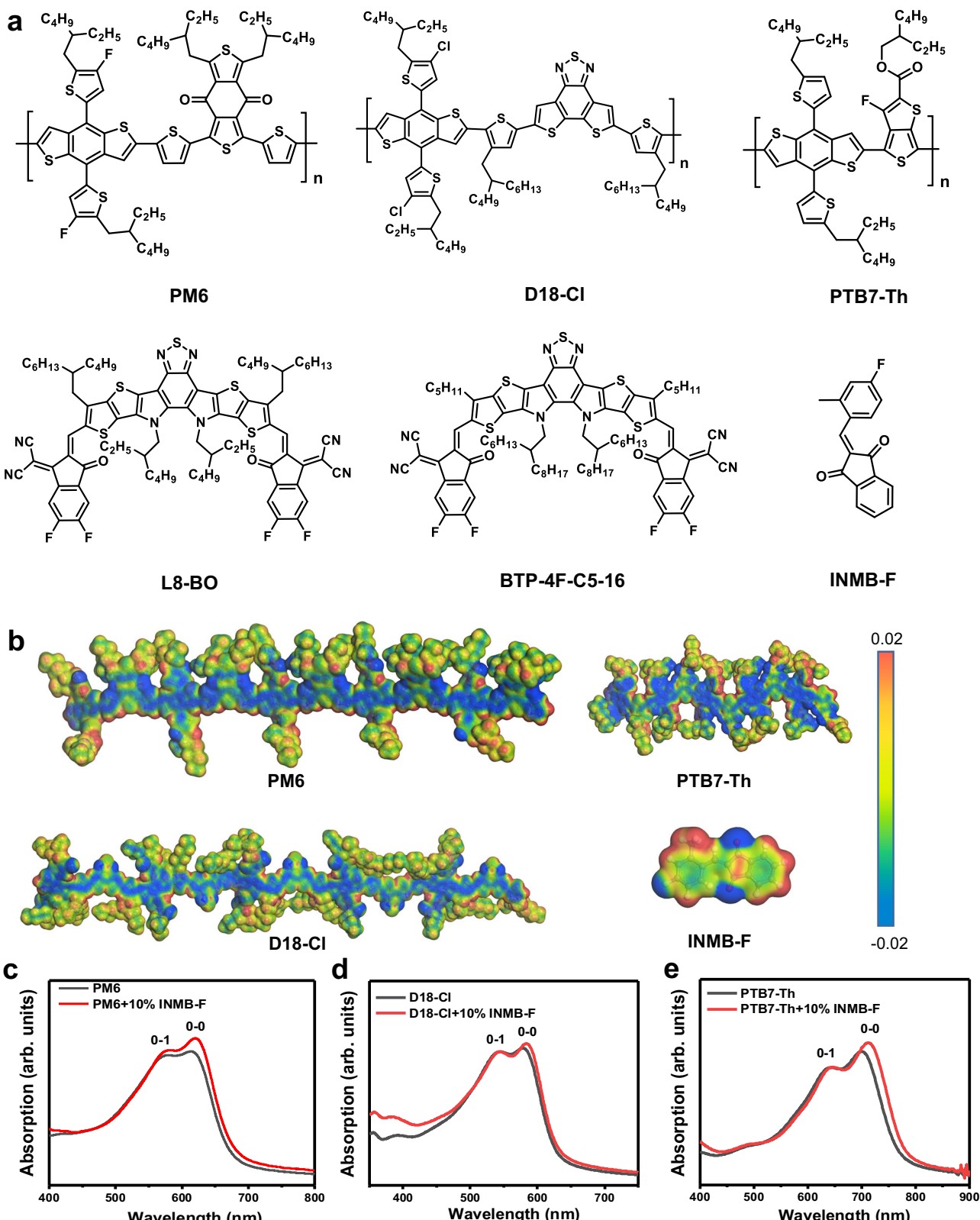

**Fig. 1 | Chemical structures, electrostatic potential and absorption spectra of materials. a** Chemical structures of PM6, D18-Cl, PTB7-Th, L8-BO, C5−16 and INMB-F, **b** Electrostatic potential of INMB-F, PM6, D18-Cl and PTB7-Th, **c**−**e** UV−vis absorption spectra of PM6, D18-Cl and PTB7-Th neat films fabricated with or without the presence of INMB-F.

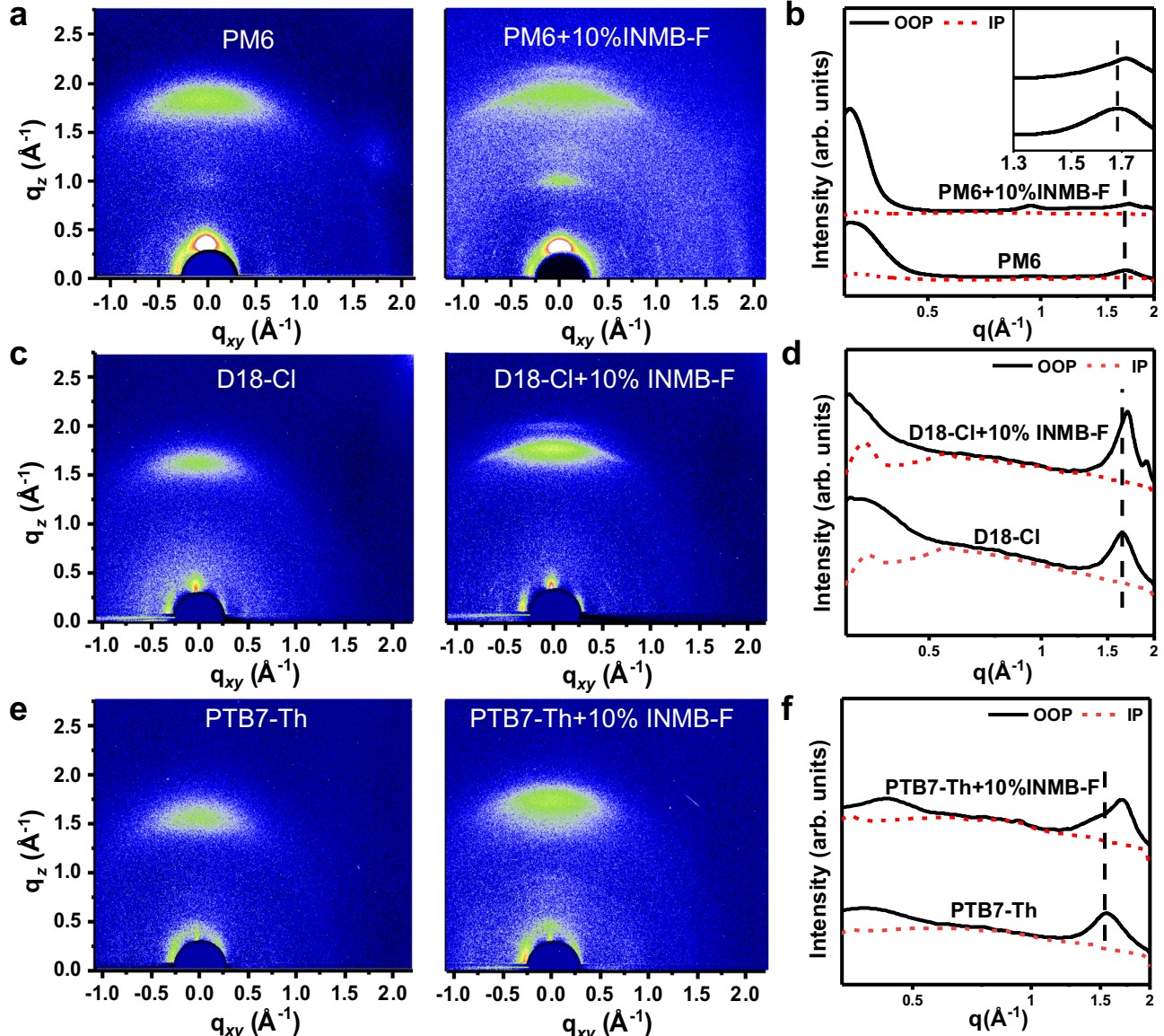

**Fig. 2 | The structural order of polymer donors. a, b** 2D GIWAXS patterns and corresponding 1D GIWAXS profiles of PM6 with and without the presence of INMB-F, **c, d** 2D GIWAXS patterns and corresponding 1D GIWAXS profiles of D18-Cl with and without the presence of INMB-F, **e, f** 2D GIWAXS patterns and corresponding 1D GIWAXS profiles of PTB7-Th films with and without the presence of INMB-F.

and from 3.83 to 3.70 Å respectively, verifying the universality of INMB-F to increase the structural order of BDT-based polymeric semiconductors in OSCs. To the best of our knowledge, this extraordinarily shortened π-π stacking distance was generally realized via chemical structure design but not through physical approaches, and hence we believe our method could offer more flexibilities in the molecular control of various organic semiconductors towards favorable optoelectronic properties, e.g. light absorption, exciton dissociation and charge transport. The hole mobilities ($\mu_e$) of three donors are measured by space-charge-limited current (SCLC) method. As shown in Supplementary Fig. 16 and Supplementary Table 2, the $\mu_h$ of PM6, D18-Cl and PTB7-Th were improved from $5.6 \times 10^{-4}$, $5.3 \times 10^{-4}$, $4.6 \times 10^{-4}$ to $8.2 \times 10^{-4}$, $6.6 \times 10^{-4}$, $6.0 \times 10^{-4}$ cm$^2$ V$^{-1}$ s$^{-1}$, respectively with the addition of INMB-F, indicating that the reduced packing distance can enhance charge transport.

**Molecular dynamics simulations**

Molecular dynamics simulations (MDS) were further employed to understand how INMB-F interacts with polymer donors and affect their structural order in solid films. From the snapshots of the simulation results shown in Fig. 3, we found that INMB-F molecules preferentially locate near the main chain of polymer donors, consistent with the above electrostatic potential surface results shown in Fig. 1. Then, by taking PM6 as an example, PM6 molecule containing 5 repeating units was selected as a substrate to allow another PM6 molecule to adsorb on, and the corresponding interaction energy (E) between two PM6 donor molecules, without (i.e. $E_{D-D}$) or with (i.e. $E_{D-D}'$) the presence of INMB-F were calculated and shown in Fig. 3. The lower the $E_{D-D}$ value, the stronger the intermolecular interaction[47]. Without the assistance of INMB-F, the calculated $E_{D-D}$ is of around -246 kcal/mol, and this value can go more negative to around -403 kcal/mol with the addition of INMB-F, confirming INMB-F could act as a molecular bridge to strengthen the interactions between adjacent polymer donors to form more compact π−π stacks. Same as PM6, we found both PTB7-Th and D18-Cl also show a reduced interaction energy from -144 and -150 kcal/mol to -387 and -263 kcal/mol, respectively (Fig. 3), further validating the versatility of the function of INMB-F which acts as molecular bridge to construct compact and ordered molecular packing of polymer donors.

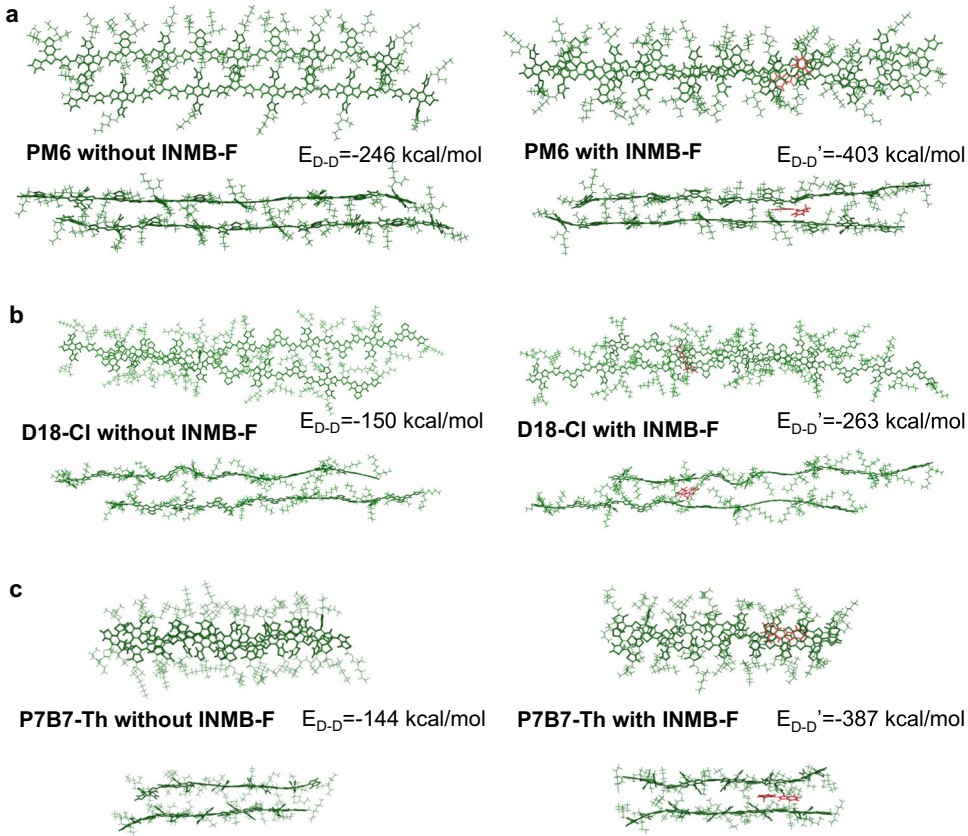

**Fig. 3 | Molecular dynamics simulations. a–c** Snapshot of molecular dynamics simulation showing intermolecular interactions between PM6, D18-Cl and PTB7-Th without and with the presence of INMB-F. The donor-donor interaction energy without INMB-F ($E_{D-D}$) and with INMB-F ($E_{D-D}'$) are also shown.

## Photovoltaic performance of OSCs

Then OSCs with a conventional device structure of indium tin oxide (ITO)/ poly(3,4-ethylenedioxythiophene) polystyrene sulfonate (PEDOT:PSS)/active layer/2,9-bis(3-((3- (dimethylamino)propyl)amino)propyl)anthra[2,1,9-*def*:6,5,10-*d'e'f'*]diisoquinoline-1,3,8,10(2*H*,9*H*)-tetraone (PDINN)/Ag were fabricated by sequential deposition of the NFA layer on the INMB-F modified donor layer to form pseudo bulk heterojunction OSCs[47]. We first fabricated PM6/C5-16 devices with different amounts of INMB-F, and the corresponding photovoltaic performance are shown in Supplementary Fig. 17 and summarized in Supplementary Table 3, from which it can be see that the device with 10% of INMB-F reached a superior PCE of 18.5% with the FF of 80.2%, $J_{SC}$ of 27.65 mA cm$^{-2}$, compared to the control device with PCE of 17.8%, FF of 78.5% and $J_{SC}$ of 27.18 mA cm$^{-2}$. To clarify the effect of the volatilization of INMB-F on photovoltaic performance, PM6/BTP-4F-C5-16 OSCs with and without INMB-F but upon different annealing temperatures were also fabricated (see Supplementary Fig. 18 and Supplementary Table 4). It is clear to see that PM6/BTP-4F-C5-16 devices with or without INMB-F follows the same temperature dependence but the one with INMB-F decreases more abruptly at the high temperature region (100 to 150 °C), where INMB-F starts to volatilize.

Subsequently, INMB-F was further utilized in other polymer:NFA systems including PTB7-Th/C5-16, D18-Cl/L8-BO and PM6/L8-BO, with results summarized in Table 1. As expected, all OSCs obtained distinct performance improvements (Fig. 4), mainly attributing from the increased $J_{SC}$ and FF. Among them, the PM6/L8-BO device obtained a remarkable PCE of 19.4% with FF of 81.3% and $J_{SC}$ of 26.94 mA cm$^{-2}$, which is one of the highest reported efficiencies of this material system[39,48] (see literature survey in Supplementary Table 5, Supporting Information). This champion device obtained a certified PCE of 18.96% at the National Photovoltaic Product Quality Inspection & Testing Center (China) (Supplementary Fig. 19). The corresponding external quantum efficiency (EQE) spectra are shown in Fig. 4f, from which enhanced photon-to-electricity response from 500 to 700 nm can be observed and the error of the integrated current density values and $J_{SC}$ obtained from $J–V$ is within 5%, proving the reliability of devices performance. The more balanced and faster hole and electron mobilities endowed by INMB-F contribute to these higher FF and $J_{SC}$ (Supplementary Fig. 20 and Supplementary Table 6).

To further investigate the improved performance of OSCs, the exciton dissociation efficiency ($P_{diss}$) and charge collection efficiency ($P_{coll}$) are calculated and summarized in Supplementary Fig. 21 and Supplementary Table 7 according to previous work[45,47]. We found that OSCs with INMB-F show negligible change in their $P_{diss}$ but decent improvement in their $P_{coll}$, suggesting that the ordered molecular packing and shortened π-π stacking preferentially optimize the charge collection process of OSCs. Meanwhile, the dependences of $J_{SC}$ and $V_{OC}$ on light intensity ($P_{light}$) were further investigated and shown in Supplementary Fig. 22. The slopes ($α$) of all devices are close to unity, suggesting that all OSCs possess relatively weak bimolecular recombination[26]. The slopes ($S$) of devices upon INMB-F modification are all smaller than the control devices, indicating reduced trap-assisted recombination[35]. Additionally, transient photovoltage (TPV) and transient photocurrent (TPC) further show that all the devices with INMB-F possess longer carrier lifetime and shorter carrier extraction time, validating the effectiveness of more ordered molecular packing and shortened π–π stacking distance for efficient photovoltaic process (Supplementary Fig. 23).

To gain deeper insights into the morphology evolution of the active layer upon INMB-F, AFM, GIWAXS and grazing incident small angle X-ray scattering (GISAXS) with or without INMB-F were conducted (Supplementary Figs. 24-26), employing PM6/L8-BO films as

**Table 1 | Summary of device parameters of different OSCs**

| Photovoltaic system | PCE (%) | FF (%) | $J_{SC}$ (mA cm$^{-2}$) | $J_{SC}$ cal. (mA cm$^{-2}$) | $V_{OC}$ (V) |
|---|---|---|---|---|---|
| PM6/C5-16 | 17.8 (17.6 ± 0.2) | 78.5 (77.8 ± 0.8) | 27.18 (26.99 ± 0.17) | 26.21 | 0.835 (0.834 ± 0.002) |
| PM6 + 10% INMB-F/C5-16 | 18.5 (18.2 ± 0.3) | 80.2 (79.5 ± 0.6) | 27.65 (27.45 ± 0.17) | 26.58 | 0.834 (0.833 ± 0.001) |
| PM6/L8-BO | 18.4 (18.2 ± 0.2) | 78.9 (78.0 ± 0.7) | 26.56 (26.40 ± 0.11) | 25.67 | 0.882 (0.879 ± 0.004) |
| PM6 + 10% INMB-F/L8-BO | 19.4 (19.1 ± 0.2) | 81.3 (80.6 ± 0.5) | 26.94 (26.8 ± 0.16) | 25.93 | 0.883 (0.881 ± 0.003) |
| D18-Cl/LB-BO | 17.3 (17.1 ± 0.2) | 76.5 (75.9 ± 0.6) | 24.87 (24.50 ± 0.23) | 23.95 | 0.909 (0.907 ± 0.002) |
| D18-Cl+10% INMB-F/LB-BO | 17.8 (17.5 ± 0.2) | 77.5 (77.1 ± 0.5) | 25.38 (24.81 ± 0.31) | 24.42 | 0.907 (0.906 ± 0.002) |
| PTB7-Th/C5-16 | 11.2 (11.1 ± 0.1) | 67.8 (67.5 ± 0.3) | 25.11 (24.80 ± 0.32) | 24.20 | 0.659 (0.656 ± 0.002) |
| PTB7-Th+10% INMB-F/C5-16 | 11.8 (11.5 ± 0.2) | 69.1 (68.8 ± 0.4) | 25.75 (25.01 ± 0.56) | 24.76 | 0.660 (0.659 ± 0.002) |

Note: average values with standard deviation were obtained from 16 individual devices.

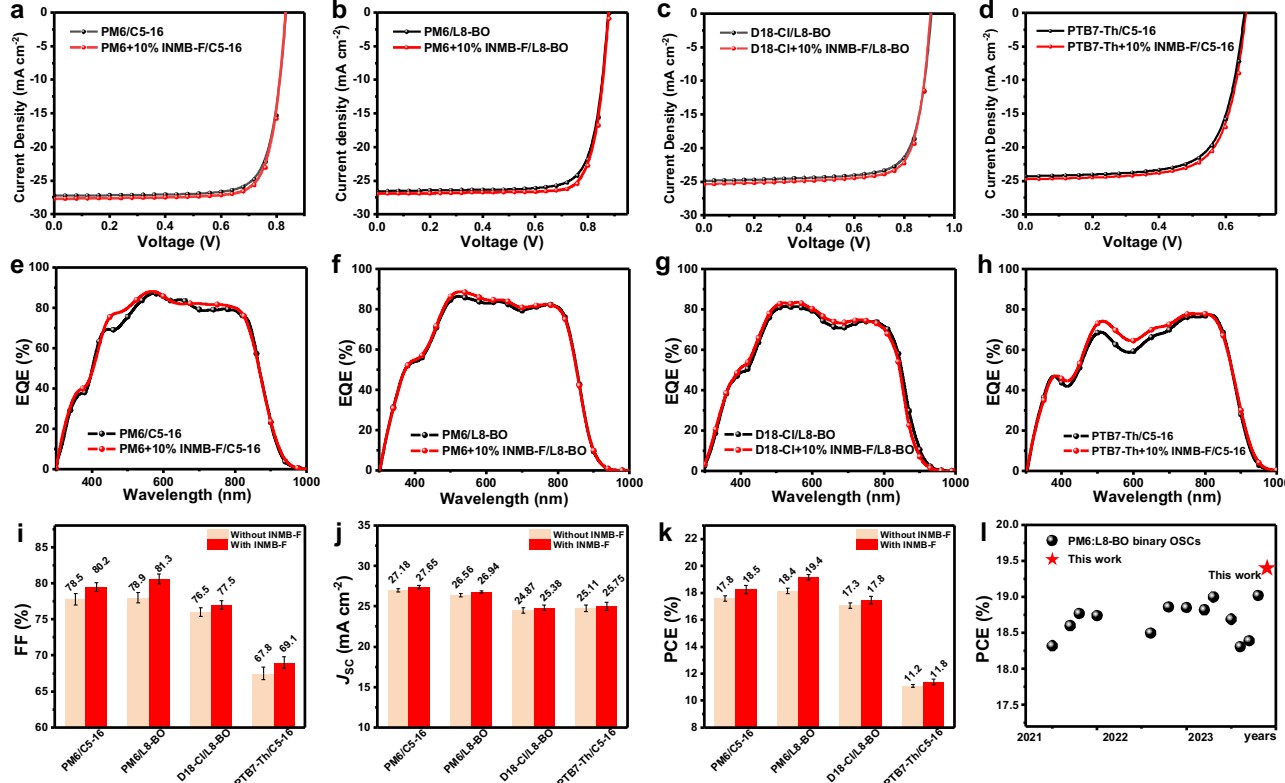

**Fig. 4 | Photovoltaic performance of OSCs. a–d** *J–V* curves of OSCs based on PM6/C5-16, PM6/L8-BO, D18-Cl/L8-BO and PTB7-Th/C5-16, **e–h** EQE spectra of OSCs based on (**e–h**) PM6/C5-16, PM6/L8-BO, D18-Cl/L8-BO and PTB7-Th/C5-16, **i–k** average FF, $J_{SC}$, and PCE values with scale bars, with the highest values presented at the top, **l** Summary of PCEs for PM6:L8-BO binary OSCs from the literatures and this work.

the example. Same as the neat PM6 film, the blend film also shows an enhanced (010) π-π stacking peak in the OOP direction with q values of π-π stacking changing from 1.72 Å$^{-1}$ to 1.74 Å$^{-1}$, translating to a *d*-spacing reducing from 3.65 to 3.6 Å. GISAXS (with analysis details shown in supplementary information and fitting parameters summarized in Supplementary Table 8[49]) and AFM results further indicate that the improved crystallinity and ordered structure of the active layer have also enlarged polymer and acceptor domains from 15.6 to 18.3 nm and 13.7 to 20.6 nm, respectively, resulting in slightly increased phase separation between donor and acceptor which is favorable for charge transport. To further clarify if INMB-F could move to the NFA phase and affect its molecular packing during device fabrication, we also made INMB-F/NFA films by spin-coating INMB-F on substrate that is followed by the casting of NFA on INMB-F. As shown in Supplementary Fig. 27, negligible changes have been observed in their absorption

spectra, indicating that INMB-F that has moved to the NFA phase during thermal annealing won't affect the aggregation of NFAs.

Previous works have reported that the existence of either solid or solvent additives in OSCs could reduce device stability[33,50]. We further evaluated the operational stability of encapsulated PM6/L8-BO devices with or without INMB-F upon continuous light illumination (white light-emitting diode with intensity of 100 mW cm$^{-2}$) in the atmospheric environment at a room temperature of 20 °C. As shown in Supplementary Fig. 28, the PCE degradation of OSCs is mainly attributed to decreased FF and $V_{OC}$, but the degradation is slowed down with the presence of INMB-F, further confirming that the increased structural order of polymer donors and tighter π−π stacking distance might be able to prohibit intermixing between donor and acceptor, therefore maintaining decent domain purity for the active layer which corresponds to enhanced morphological and operational stability. As the

results, the operational stability of PM6/L8-BO device processed with INMB-F obtained a superior $T_{80}$ of 600 h, six times higher than that of the device processed without INMB-F (100 h).

In conclusion, we report a conjugated molecule INMB-F, which can interact with the conjugated main chains of a range of BDT-based polymer donors via electrostatic force (dipole-dipole interaction), and drive polymer donors to obtain stronger intermolecular interactions. As the results, INMB-F not only leads to a significantly increased structural order with reduced π-π stacking distance among donors, but also stabilized the active layer to avoid donor/acceptor intermixing. As such, INMB-F brings generally enhanced photovoltaic performance in PM6/C5-16, PTB7-Th/C5-16, D18-Cl/L8-BO and PM6/L8-BO OSCs, with a PCE of 19.4% achieved in PM6/L8-BO OSC as well as sixfold improvement of the operational stability in air. This work demonstrates a new approach to enhance the structural order of organic semiconductors via electrostatic force to improve photovoltaic performance.

## Methods

### Materials
PM6 ($M_w$ = 97781 Da, PDI = 2.4), D18-Cl ($M_w$ = 71084 Da, PDI = 2.1), PTB7-Th ($M_w$ = 40000 Da, PDI = 2.0) and PDINN were purchased from Solarmer Materials (Beijing) Inc. BTP-4F-C5-16 was synthesized in our previous work[51]. The synthesis and characterization details of INMB-F are shown in the following section. Other chemicals, unless otherwise specified, were purchased from commercial resources and used as received. High-transmittance ITO-glass substrates (resistance ~12 Ω sq⁻¹, maximum transmittance ~94% at ~550 nm, size of 20 × 15 × 0.7 mm³) were purchased from You Xuan Ltd. China. The synthesis details of INMB-F were provided in the supporting information.

### Solar cell device fabrication
All solar cell devices were fabricated with a conventional structure (ITO/PEDOT:PSS/Donors/ NFAs/PDINN/Ag). Pre-patterned ITO-glass substrates were cleaned by sequential sonication in deionized water, ethanol, and isopropyl alcohol for 10 min each before drying at 120 °C on a hotplate. Subsequently, they were treated with ultraviolet/Ozone for 20 min. Then PEDOT:PSS aqueous solution was spin-coated at 5000 rpm on the top of cleaned ITO substrates and annealed at 150 °C for 15 min in air to allow thickness of *ca.* 15 nm. For the layer-by-layer architecture of PM6/acceptors, the PM6 layer was firstly spin-coated from 7 mg/ml CF solution at 2000 rpm for 30 s, and the acceptor layer was deposited on the donor layer from an 8 mg/ml CF solution at 3000 rpm for 30 s, the optimal thickness of the active layer was about 95 nm. For the layer-by-layer architecture of D18-Cl/L8-BO, the D18-Cl layer was firstly spin-coated from 6 mg/ml CF solution at 2000 rpm for 30 s, and the L8-BO layer was deposited on the D18-Cl layer from an 8 mg/ml CF solution at 3000 rpm for 30 s, the optimal thickness of the active layer was about 95 nm. For the layer-by-layer architecture of PTB7-Th/C5-16, the PTB7-Th layer was firstly spin-coated from 8 mg/ml CF solution at 2500 rpm for 30 s, and the C5-16 layer was deposited on the PTB7-Th layer from an 8 mg/ml CF solution at 2500 rpm for 30 s, the optimal thickness of the active layer was about 100 nm. The as-cast films were then thermally annealed at 80 °C for 5 min to fully evaporate CF. To investigate the volatility of INMB-F, INMB-F solution was spin-coated on glass substrate, and then thermally annealed at different temperatures of 80 °C, 100 °C and 120 °C to observe the amount of residual INMB-F and measure the corresponding thickness. A thin layer (~10 nm) of PDINN was spin-coated (3000 rpm for 30 s) on the top of the active layer from 1.5 mg/ml methanol solution. Finally, 100 nm Ag was thermally evaporated under high vacuum through the shadow mask to form the anode. Each single-substrate device consists 4 individual pixels that each can operate separately, and the active area of each pixel is 6.625 mm² defined by the overlapping of anode and cathode.

### Instruments and measurements
A Bruker Avance III HD 500 MHz spectrometer was used to measure the ¹H, ¹³C and ¹⁹F NMR spectra of all compounds. Film absorption spectra were carried out using a UV-visible spectrophotometer (HITACHI, Japan). The current density–voltage (J–V) measurements were performed under AM 1.5 G (100 mW cm⁻²) using a Newport 3 A solar simulator (Newport, USA) in air at room temperature after the light intensity was calibrated using a standard silicon reference cell certified by the National Renewable Energy Laboratory (NREL, USA). J–V characteristics were measured using software developed by Ossila Ltd. (UK) together with a source meter unit (2612B, Keithley, USA). An aperture mask was placed over the device to define an accurate illumination area of 4.04 mm² for each pixel. External quantum efficiency (EQE) was measured with an EQE system (Zolix, China) equipped with a standard Si diode. Water contact angle measurement system (Attension Theta Lite), and the surface energy was calculated using the equation of state. The surface morphology of all films was characterized by an atomic force microscope (AFM) (Solver Next, NT-MDT, Russia). Grazing-incidence wide-angle X-ray scattering (GIWAXS) and grazing-incidence small-angle X-ray scattering (GISAXS) measurements were conducted using the beamline BL14B1 and BL16B1 at the Shanghai Synchrotron Radiation Facility in China.

### Calculation of Herman's orientation parameter
The 100-diffraction peak is located at around 0.3 Å⁻¹, so we take the *q*-χ figure by choosing the range of *q* from 0.25 to 0.40 Å⁻¹ of PM6 films and from 0.25 to 0.36 Å⁻¹ of PM6 film with INMB-F (polar angles 0°–2° are not accessible in the GIWAXS geometry at the wavelength used). We have calculated the Herman's orientation parameter (*S*) of the 100-diffraction peak via equation[46] below:

$$f_\perp = \frac{\int_0^{\pi/2} I(\chi)\cos^2(\chi)\sin(\chi)d\chi}{\int_0^{\pi/2} I(\chi)\sin(\chi)d\chi} \qquad (1)$$

I(χ) is the total scattered intensity that was determined as the area of the Gaussian of the (100) peak at each polar angle and the sin(χ) term is a geometric intensity correction factor.

$$S = \frac{1}{2}(3f_\perp - 1) \qquad (2)$$

The S value ranges from -0.5 to 1, where -0.5 corresponds to all lattice plane oriented perpendicular to the substrate normal while 1 corresponds to all lattice plane oriented parallel to the substrate normal.

### Molecular dynamics simulations
The optimized geometry of INMB-F and PM6, D18-Cl and PTB7-Th (simplified with 5 repeat units) were obtained using the Forcite and DMol³ modules. Compass II forcefield was used to assign charges in the Forcite module and the atom-based summation method was used to determine the van der Waals interactions. The results from Forcite were performed to obtain geometry optimization in the DMol³ module through the Perdew−Burke−Ernzerhof (PBE) and generalized gradient approximation (GGA) functions, with the basis set as DNP. The electrostatic potential (ESP) mapped figures and torsion angles of additive and donors were obtained using energy task in the DMol³ module. The optimized INMB-F and donors molecules calculated from DMol³ were put in the adsorption locator module. We put two donor molecules as the adsorbate and one INMB-F molecule as the substrate to determine their interactions and aggregation. The geometry optimization and energy tasks in the Forcite module were used to gain the interaction energy (i.e. adsorption energy in the software) between donor and INMB-F[33].

## Transient photovoltage (TPV) and transient photocurrent (TPC) measurements

The transient photocurrent and transient photovoltage characteristics of devices were measured by applying 405 nm laser square pulse with rise and fall times under 5 ns. The charge extraction time and charge carrier lifetime were extracted from the fitting line of the TPC/TPV signal with the equation:

$$y = a^* e^{\left(-\frac{x}{\tau}\right)} + c \qquad (3)$$

where a is a constant that fits the peak high, $x$ is time, $\tau$ is corresponding to the charge extraction time (TPC), while $\tau$ is corresponding to the charge carrier lifetime (TPV).

## Charge mobility measurements

The electron-only device with the structure of ITO/ZnO/Active layer/PDINN/Ag and the hole-only device with the structure of ITO/PEDOT:PSS/Active layer/MoO$_3$/Ag were fabricated. The SCLC measurements employ the Mott-Gurney equation: $J = 9\varepsilon_0\varepsilon_r\mu V^2/8L^3$ to estimate the electron and hole mobilities from dark $J$-$V$ curves obtained from these devices. Here, $J$ is the current density, $\varepsilon_r$ is the relative dielectric constant of the active layer, $\varepsilon_0$ is the permittivity of free space, $\mu$ is the charge mobility, and $L$ is the thickness of the active layer. $V = V_{app} - V_{bi}$, where $V_{app}$ is the voltage applied to the OSC device, and $V_{bi}$ is the built-in potential voltage.

## GISAXS modeling

To quantify and compare the phase separation in the PM6/L8-BO photovoltaic blends, the 1D GISAXS profiles were fitted using a DAB +Fractal model expressed in Equation 4 via the SASView (Version 4.2.2). The first term of the equation is Debye-Anderson-Brumberger (DAB) model, where q is the scattering wave vector, A$_1$ is an independent fitting parameter, and ξ is the average correlation length of the polymer domain. The second term of the equation is the Fractal model, describing the occupation of fractal-like structure of the non-fullerene acceptor. P (q, R) and S (q, R) are the form and structure factors, respectively. The correlation length of the fractal-like structure is represented by $\eta$ and D is the fractal dimension.

$$I(q) = \frac{A_1}{\left[1 + (q\xi)^2\right]^2} + A_2(P(q,R))S(q,R,\eta,D) + B \qquad (4)$$

## Reporting summary

Further information on research design is available in the Nature Research Reporting Summary linked to this article.

## Data availability

All the data that support the findings of this study are available within the main text and Supplementary Information file, and also available from the corresponding author on request. Source data are provided with this paper.

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

## Acknowledgements

This work is supported by the National Natural Science Foundation of China (52273196, 52073221 and 52203238), and the Key Research and Development Program of Hubei Province (2023BAB116). We thank beamline BL14B1 and BL16B1 at Shanghai Synchrotron Radiation Facility (China) for providing beamtime to perform GIWAXS and GISAXS measurements.

## Author contributions

Z.G., D.L., L.W., and C.G. fabricated and optimized devices. Z.G. performed J–V, $J_{ph}$-$V_{eff}$, EQE, UV–vis, AFM, Plight versus $J_{SC}$ and $V_{OC}$ plots, TPC, TPV, Raman spectra and water contact angle measurements. J.C. synthesized INMB-F. L.W. synthesized C5-16. C.G. prepared hole-only and electron-only devices. C.C. and Z.G. performed GIWAXS, GISAXS, ESP and MDS. Z.G., L.W. and B.Z. performed stability measurements. Y.F. helped with UV–vis measurements. Y.S. helped with water contact angle measurements. C.L. helped with TPC and TPV measurements. J.Z. and D.L. helped with AFM measurements. Z.G., W.L., and T.W. wrote the manuscript with all authors commented and revised this paper. This project was supervised by T.W.

## Competing interests

The authors declare no conflict of interest.
