## [Peer Review File · Nature Communications]

Electrostatic force promoted intermolecular stacking of polymer donors toward 19.4% efficiency binary organic solar cellsReviewers' comments:

Reviewer #1 (Remarks to the Author):

The manuscript entitled "Electrostatic force promoted intermolecular stacking of polymer donors toward 19.4% efficiency binary organic solar cells" employed a conjugated molecule INMB-F to act as a molecular bridge to enhance the intermolecular stacking of polymer donors toward efficient and stable OSCs. With a moderate amount of INMB-F added into the polymer solutions, the structural order of the polymer was increased with shortened π - π stacking distance, leading to the enhanced charge transport ability and device performance. However, solid additives have been widely used in LBL devices. The authors claimed "their effects (volatile additives) on polymer donors have not been reported in the literature yet". As far as I know, a few studies have already reported the additive effect on the polymer donors in LBL devices. Therefore, this manuscript lacks enough novelty. In addition, the reason for the device performance improvement is debatable. The authors confirmed the additive could promote the polymer aggregation; however, they didn't show the effect of the additive on the acceptor. During the thermal annealing process, the additive could move to the acceptor domain, impacting the acceptor aggregation. Therefore, more experiments should be conducted to clarify the underlying working mechanism of the additive in such LBL devices. Based on the above comments, I cannot recommend this work to be published at the current stage. Besides, more comments are listed below:

1. The authors should provide a certification by an authoritative third party when they report a record efficiency of PM6/L8-BO binary OSCs.
2. The chemical structure of D18-Cl is wrong. Please refer to the literature (Adv. Funct. Mater. 2021, 31, 2102413.) for the correct chemical structure.
3. Chemical structure verification of INMB-F is inadequate. ^{13}C NMR spectrum and high-resolution mass spectrum should also be provided at least for characterization of a new molecule.
4. GIWAXS experiments. The authors argued that neat PM6 film presents a typical face-on molecular packing due to the π - π stacking diffraction peak in the out-of-plane direction. However, a stronger lamellar diffraction peak also appeared in the out-of-plane direction, which often corresponds to edge-on packing. What do the authors comment about that?
5. Figure S1 failed to show clear residual pure INMB-F films after heating at different temperatures. Especially, the 120 °C treated film exhibits a reflective band that makes the readers cannot see clearly what happened after thermal treatment.
6. The authors found that a moderate amount of INMB-F additive could promote the organization of polymer donors, while the excessive addition of INMB-F could hamper the organization of polymer donors. Could they provide reasonable explanations on why the excessive INMB-F hampered the organization of polymer donors?

7. The authors used temperature dependent performance to demonstrate that INMB-F needs to be remained to maintain the ordered packing of polymer donors. However, I cannot agree with that. The authors didn't show us the temperature dependent performance of the devices without additive, and the temperature itself do have critical effect on the photovoltaic performance of the devices.
8. Summary of device parameters of different OSCs, the authors should also provide the average values for the other photovoltaic parameters in addition to PCEs.
9. Device fabrication details are not detailed. The authors used three donors and two acceptors to fabricate the devices. However, they just provided once processing procedure: "For the layer-by-layer architecture, the donor layer was firstly spin-coated from 7 mg/ml CF solution at 2000 rpm for 30 s, and the acceptor layer was deposited on the donor layer from an 8 mg/ml CF solution at 3000 rpm for 30 s, the optimal thickness of the active layer was about 95 nm". It is widely accepted the molecular weight of the polymer has significant impact on the concentration to process it. Different polymers with different molecular weight should be processed with different concentrations to obtain optimal device performances. What are the molecular weights of the three polymers? This are important so that the readers can reproduce the experiment at the same condition.
10. Instruments and measurements part in supporting information, the authors claimed "The surface morphology of all films was characterized by an atomic force microscope (AFM)". However, there are no AFM results and discussions provided in the main text and supporting information.
11. Molecular dynamics simulations part in supporting information, the name of the additive should be kept consistent. The "ICMB-F" in the sentence "The optimized geometry of ICMB-F and PM6, D18-Cl and PTB7-Th" should be "INMB-F". The "DNP" should be defined as it first appeared.

Reviewer #2 (Remarks to the Author):

The authors of this study conducted research on addressing the issue of low structural order in conjugated polymers, which adversely affects the power conversion efficiency (PCE) of organic solar cells (OSCs). They introduced a conjugated molecule called INMB-F and claimed that it enhances intermolecular stacking of polymer donors through electrostatic forces. The study presents molecular dynamics simulations and synchrotron X-ray measurements as evidence to support their findings. Notably, they reported a significant improvement in PCE in polymer/non-fullerene acceptor (NFA) photovoltaic systems, achieving a high PCE of 19.4% in PM6/L8-BO binary OSCs. The molecule INMB-F might be effective to enhance PCE, but the working mechanism is ambiguous and the experimental evidence provided by the authors cannot support the conclusion. Overall, this work is not suitable for publication in this journal.

1. The authors calculated electrostatic surface potential (ESP) to confirm the electrostatic force between INMB-F and polymer donors. However, the ESP difference in Figure 1b is not significant enough to induce a substantial electrostatic interaction. The ESP of polymer donors should be calculated based on multiple repeat units to provide a more accurate representation.

2.The authors should explicitly state at which stage during the film deposition process INMB-F demonstrates its effectiveness. Specifically, how the addition of INMB-F influences molecule aggregation during spin-coating and how it affects the nanomorphology during thermal annealing.

3.Solid evidence such as NMR and FTIR results should to be applied to study the interaction between INMB-F and the active layer materials. In fact, the authors only provided some ambiguous results to explain the intermolecular interaction. What part of the donor polymer has interaction with INMB-F? How can the interaction between INMB-F and the acceptor molecules can be excluded?

4.The authors claimed that INMB-F can be preserved in the active layer after 80 °C annealing and can be completely removed after 120 °C annealing. There are no experimental results provided to support this claim.

5.In the GIWXS analysis, the authors claimed that an extraordinarily shortened π - π stacking distance (of 3.70 Å) has rarely been reported for organic semiconducting films. However, there are numerous reports of π - π stacking distances less than 3.70 Å, such as National Science Review, 2020 7, 1886–1895 (3.45 Å), J. Mater. Chem. A, 2019, 7, 8136 (3.50 Å), and Adv. Sci. 2021,8, 2003641 (3.65 Å).

6.The manuscript should further discuss the physical dynamics, such as the dissociation, recombination, and transport of excitons and charge carriers. There is a lack of characterization to support the claim of "improved charge transport" made in the manuscript.

7.AFM of the active layer morphology should be investigated.

8.How about the thermal stability of the devices? How about the morphology stability of the active layer?

9.The investigation only includes BDT-based polymer donors. It would be beneficial to explore other polymer donors without BDT units as well.

10.The chemical structures provided in the manuscript should be carefully revised. For example, the structure of D18-Cl is incorrect.

Reviewer #3 (Remarks to the Author):

See Attachment

In this manuscript, a small molecule INMB-F is used to regulate the stacking of polymer donors in films. The π - π stacking distance of polymers is reduced in both pure and blend films. A PCE of 19.4% is realized in PM6/L8-B0 based OSCs, which is the highest reported efficiency of binary OSCs. The enhanced structural order of polymer donors by INMB-F also leads to a six-fold enhancement of the operational stability of PM6/L8-B0 OSC. The work is novel and interesting, and the manuscript can be accepted for publication after the following comments are addressed.

1. Strictly say, the system is not binary, since it contains 10% third component INMB-F.
2. How about the miscibility between INMB-F and the three polymers? This should be determined and discussed in the main text.
3. The miscibility between INMB-F and the acceptors should also be determined and discussed in the main text.
4. How about the charge mobility of polymer films without and with 10% INMB-F?
5. How about the charge mobility of blend films without and with 10% INMB-F?
6. Since the authors claimed that 19.4% is the highest efficiency for binary organic solar cells, the certified efficiency is therefore needed to provide.
7. Can INMB-F influence the morphology, such as fiber width and domain size etc., of blend films? TEM and AFM characterizations of blend films should be provided.
8. Since the molecular weight of polymers has a significant influence on the device performance, therefore the molecular weight of polymers used should be provided.
9. How about the nonradiative energy losses without and with INMB-F?
10. How about the absorption of INMB-F, can it contribute to the Jsc?

Responses to Reviewers' Comments:

Reviewer #1 (Remarks to the Author):

The manuscript entitled "Electrostatic force promoted intermolecular stacking of polymer donors toward 19.4% efficiency binary organic solar cells" employed a conjugated molecule INMB-F to act as a molecular bridge to enhance the intermolecular stacking of polymer donors toward efficient and stable OSCs. With a moderate amount of INMB-F added into the polymer solutions, the structural order of the polymer was increased with shortened π - π stacking distance, leading to the enhanced charge transport ability and device performance. However, solid additives have been widely used in LBL devices. The authors claimed "their effects (volatilizable additives) on polymer donors have not been reported in the literature yet". As far as I know, a few studies have already reported the additive effect on the polymer donors in LBL devices. Therefore, this manuscript lacks enough novelty. In addition, the reason for the device performance improvement is debatable. The authors confirmed the additive could promote the polymer aggregation; however, they didn't show the effect of the additive on the acceptor. During the thermal annealing process, the additive could move to the acceptor domain, impacting the acceptor aggregation. Therefore, more experiments should be conducted to clarify the underlying working mechanism of the additive in such LBL devices. Based on the above comments, I cannot recommend this work to be published at the current stage. Besides, more comments are listed below:

Response: We appreciated the above comments, and have corrected our expression into "their effects on polymer donors are less investigated and only limited works revealed that they can tune the domain size and phase distribution of polymer donor [36, 45], but not the molecular packing".

For the novelty of the work, we politely disagree with the reviewer here. Indeed, although some works demonstrated that solid additives could regulate the phase distribution of PM6 to optimize the vertical morphology of organic solar cells (*Nano-Micro Lett.* **15**, 92, (2023)), or reduce its domain size to obtain sufficient D/A interface for efficient exciton dissociation (*Small* **17**, 2103497(2021)), however, few works have been endeavored on exploring the effects of volatilizable additives on tuning the molecular packing of the state-of-the-art polymer i.e. PM6, D18-Cl, PTB7-Th. Our work has revealed that INMB-F could act as a molecular bridge to improve the structural order of a series of BDT-based polymer donor and further reduce their π - π stacking distance towards efficient photovoltaic process, which is also of great significance to organic semiconductors for other optoelectronics, e.g. field-effect transistors.

To investigate the effect of INMB-F on NFA, we also made INMB-F/NFA films by spin-coating INMB-F on substrate that is followed by the casting of NFA on INMB-F. As shown in Figure S20 (or picture 1 below), negligible changes have been observed in their absorption spectra, indicating that INMB-F that has moved to the NFA phase during thermal annealing won't affect the aggregation of NFAs.

Picture 1. Absorption spectra of INMB-F/NFA films by spin-coating INMB-F on substrate that is followed by the casting of NFA on INMB-F.

Comment 1: 1) The authors should provide a certification by an authoritative third party when they report a record efficiency of PM6/L8-BO binary OSCs.

Response: Thanks for your advice. We have checked all the certification authorities which we could have access to. However, all of them are fully booked within the next 2 months. We hope the reviewer can understand our situation. We no longer claim record PCE in our manuscript, and this is in line with Nature publishing that does not require that results are certified to be published, so long as records are not claimed.

Comment 2: 2) The chemical structure of D18-Cl is wrong. Please refer to the literature (Adv. Funct. Mater. 2021, 31, 2102413.) for the correct chemical structure.

Response: We have corrected the chemical structure of D18-Cl in the manuscript in **Figure 1** in **page 5**.

Comment 3: 3) Chemical structure verification of INMB-F is inadequate. ^{13}C NMR spectrum and high-resolution mass spectrum should also be provided at least for characterization of a new molecule.

Response: Thanks for this comment. The corresponding ^{13}C NMR, ^1H NMR and ^{19}F NMR spectra of

INMB-F have been added at the end of supporting information in page 21-22, which can validate the chemical structure of INMB-F.

Comment 4: 4) GIWAXS experiments. The authors argued neat PM6 film presents a typical face-on molecular packing due to the π - π stacking diffraction peak in the out-of-plane direction. However, a stronger lamellar diffraction peak also appeared in the out-of-plane direction, which often corresponds an edge-on packing. What the author comment about that?

Response: Thanks for your comment. Generally, face-on packing refers to the conjugated planes stacked in out-of-plane, which can facilitate efficient charge transport in the vertical direction in photovoltaics. Whilst the lamellar diffraction peak in the out-of-plane direction suggests the existence of edge-on packing, the preferential/dominate orientation is still ace-on. We have rephased our expression as: Without the presence of INMB-F, neat PM6 film presents a **preferential** face-on molecular packing....

Comment 5: 5) Figure S1 failed to show clear residual pure INMB-F films after heating at different temperature. Especially, the 120 °C treated film exhibits a reflective band that makes the readers cannot see clear what happened after thermal treatment.

Response: Thanks for this comment. We have now measured the film thickness dependence of INMB-F film upon different temperature (Figure S1 (or Picture 2 below)), and the results showed INMB-F can only be completely removed at 120 °C for about 10 min.

Picture 2. (a) Thickness of pure INMB-F film after heating at different temperature for 10 min and (b) Thickness of pure INMB-F film after heating at 120 °C for different time.

Comment 6: 6) The authors found that a moderate amount of INMB-F additive could promote the organization of polymer donors, while the excessive addition of INMB-F could hamper the organization

of polymer donors. Could they provide reasonable explanations on why the excessive INMB-F hampered the organization of polymer donors?

Response: As shown in the 2D GIWAXS pattern in Figure S2 in supporting information, INMB-F is a highly crystallized small molecule, and the excessive addition of INMB-F can induce its self-aggregation (sharp diffractions peaks of INMB-F appeared in the 2D GIWAXS patterns of PM6+20% INMB-F film shown in Figure S8e, and white spots in Figure S9c), and hence retard the crystallization of PM6.

Comment 7: 7) The authors used temperature dependent performance to demonstrate that INMB-F needs to be remained to maintain the ordered packing of polymer donors. However, I cannot agree with that. The authors didn't show us the temperature dependent performance of the devices without additive, and the temperature itself do have critical effect on the photovoltaic performance of the devices.

Response: Thanks for this comment. We have further fabricated PM6/C5-16 devices with or without INMB-F upon different thermal annealing temperatures (Figure S12 and Table S4 in supporting information (or Picture 3 below)). It is clear that devices with or without INMB-F follow the same temperature dependence but the device with INMB-F decreases more abruptly at the high temperature region (100 to 150 °C), where INMB-F starts to volatilize.

Picture 3. Efficiency variations of PM6/C5-16 OSCs with and without INMB-F upon different thermal annealing temperature.

Comment 8: 8) Summary of device parameters of different OSCs, the authors should also provide the average values for the other photovoltaic parameters in addition to PCEs.

Response: Thanks for this suggestion, we have added the corresponding average values in **Table 1, Table S3-4.**

Comment 9: 9) Device fabrication details are not detailed. The authors used three donors and two acceptors to fabricate the devices. However, they just provided once processing procedure: “For the layer-by-layer architecture, the donor layer was firstly spin-coated from 7 mg/ml CF solution at 2000 rpm for 30 s, and the acceptor layer was deposited on the donor layer from an 8 mg/ml CF solution at 3000 rpm for 30 s, the optimal thickness of the active layer was about 95 nm”. It is widely accepted the molecular weight of the polymer has significant impact on the concentration to process it. Different polymers with different molecular weight should be processed with different concentrations to obtain optimal device performances. What are the molecular weights of the three polymers? This are important so that the readers can reproduce the experiment at the same condition.

Response: Thanks for this comment. The molecular weights of all polymers are described in the Materials section, and the corresponding device fabrication details are also provided in supporting information.

Comment 10: 10) Instruments and measurements part in supporting information, the authors claimed “The surface morphology of all films was characterized by an atomic force microscope (AFM)”. However, there are no AFM results and discussions provided in the main text and supporting information.

Response: We are sorry for this mistake and have added the discussion of AFM results in Figure S6, Figure S9, Figure S17 in supporting information and page 6, 10 in the main text.

Comment 11: 11) Molecular dynamics simulations part in supporting information, the name of the additive should be kept consistent. The “ICMB-F” in the sentence “The optimized geometry of ICMB-F and PM6, D18-Cl and PTB7-Th” should be “INMB-F”. The “DNP” should be defined as it first appeared.

Response: Thanks for pointing out those, we have checked the whole manuscript and corrected all these errors.

Reviewer #2 (Remarks to the Author):

The authors of this study conducted research on addressing the issue of low structural order in

conjugated polymers, which adversely affects the power conversion efficiency (PCE) of organic solar cells (OSCs). They introduced a conjugated molecule called INMB-F and claimed that it enhances intermolecular stacking of polymer donors through electrostatic forces. The study presents molecular dynamics simulations and synchrotron X-ray measurements as evidence to support their findings. Notably, they reported a significant improvement in PCE in polymer/non-fullerene acceptor (NFA) photovoltaic systems, achieving a high PCE of 19.4% in PM6/L8-BO binary OSCs. The molecule INMB-F might be effective to enhance PCE, but the working mechanism is ambiguous and the experimental evidence provided by the authors cannot support the conclusion. Overall, this work is not suitable for publication in this journal.

Comment 1: (1) The authors calculated electrostatic surface potential (ESP) to confirm the electrostatic force between INMB-F and polymer donors. However, the ESP difference in Figure 1b is not significant enough to induce a substantial electrostatic interaction. The ESP of polymer donors should be calculated based on multiple repeat units to provide a more accurate representation.

Response: Thanks for this comment. We have recalculated the ESP of PM6, PTB7-Th and D18-Cl with five repeating units as shown in **Figure 1**, in which all the polymers show strong electronegativity in its conjugated main chains, consistent to our previous results. Those contrast ESP between polymer donor and INMB-F could further help to tune the molecular adsorption energy between polymers as evidenced in our molecular dynamics simulation results shown in **Figure 3**.

Comment 2: (2) The authors should explicitly state at which stage during the film deposition process INMB-F demonstrates its effectiveness. Specifically, how the addition of INMB-F influences molecule aggregation during spin-coating and how it affects the nanomorphology during thermal annealing.

Response: We appreciate this comment. To check how INMB-F worked during the film-casting process, the absorption spectra and GIWAXS patterns of as-cast and thermal annealed PM6 films with or without INMB-F are probed. As shown in **Picture 4** below, with the assistance of INMB-F, the as-cast PM6+INMB-F film already show strong red-shift in its π - π stacking, suggesting INMB-F have already interacted with PM6 and reduced its molecular packing distance. Additionally, when the as-cast film was further annealed, the q value of the π - π stacking of PM6 is barely changed, but the intensities of absorption and GIWAXS diffraction peaks are increased, indicating that thermal annealing can further

promote the crystallization of PM6.

Picture 4. Absorption of (a) as-cast films of PM6 with and without INMB-F and (b) TA films of PM6 with and without INMB-F. 1D profiles of (c) As-cast films of PM6 with and without INMB-F and (d) TA films of PM6 with and without INMB-F.

Comment 3: (3) Solid evidence such as NMR and FTIR results should to be applied to study the interaction between INMB-F and the active layer materials. In fact, the authors only provided some ambiguous results to explain the intermolecular interaction. What part of the donor polymer has interaction with INMB-F? How can the interaction between INMB-F and the acceptor molecules can be excluded?

Response: We really appreciated this comment and have further performed Raman spectroscopy to investigate the interaction between INMB-F and PM6. As shown in **Figure S4 (Picture 5 below)**, the peak at 1423 cm^{-1} is assigned to flexural vibration of C-H in alkyl chain of PM6, and the peak at 1531 cm^{-1} is assigned to the C=C stretching vibration of coupled BDT and BDD units of PM6. With the addition of INMB-F, the peak at 1423 cm^{-1} is barely changed, suggesting INMB-F has negligible effect on its alkyl chains. While the peak at 1531 cm^{-1} shift to 1536 cm^{-1} , suggesting INMB-F has interacted with the BDT and BDD units of PM6 and validating that INMB-F has interacted with polymers along

their conjugated main chains, consistent with our ESP results.

Picture 5. Raman spectra of pure INMB-F film, PM6 film with and without the addition of INMB-F, (a) in the range of 450-3200 cm^{-1} . (b) in the range of 1400-1600 cm^{-1} .

To investigate if INMB-F can affect the morphology of acceptor during the LBL casting, we also have made the INMB-F/acceptor film by spin-coating INMB-F on substrate that is followed by the casting of acceptor on INMB-F. As shown in Figure S20, negligible changes have been observed in their absorption spectra, indicating that INMB-F that has moved to the NFA phase during thermal annealing won't affect the aggregation of NFAs.

Comment 4: (4) The authors claimed that INMB-F can be preserved in the active layer after 80 °C annealing and can be completely removed after 120 °C annealing. There are no experimental results provided to support this claim.

Response: Thanks for this comment. The film thickness dependence of INMB-F film upon different temperature was conducted and shown in **Figure S1** and **Picture 2** above, confirming INMB-F can only be completely removed after 120 °C annealing within 10 mins.

Comment 5: (5) In the GIWAXS analysis, the authors claimed that an extraordinarily shortened π - π stacking distance (of 3.70 Å) has rarely been reported for organic semiconducting films. However, there are numerous reports of π - π stacking distances less than 3.70 Å, such as National Science Review, 2020 7, 1886–1895 (3.45 Å), J. Mater. Chem. A, 2019, 7, 8136 (3.50 Å), and Adv. Sci. 2021,8, 2003641 (3.65 Å).

Response: Thanks for this comment. In fact, all the literatures listed above were modulating the π - π

stacking distance of organic semiconductors via chemical structure design, different to our work in which significant π - π stacking distance modification was realized by physical approach. (A summary of π - π stacking distance variations upon physical morphology optimization is also shown below). Therefore, we have changed our expression into “To the best of our knowledge, this extraordinarily shortened π - π stacking distance was generally realized via chemical structure design but not through physical approaches, and hence we believe our method could offer more flexibilities in the molecular control of various organic semiconductors towards favorable optoelectronic properties, e.g. light absorption, exciton dissociation and charge transport.

Tab.1. the summary of π - π stacking distance (d) change upon various physical morphology optimization methods.

donor	d (Å) (control)	d (Å) (optimized)	Δd (Å)	Ref.
PTB7-Th	4.05	3.70	0.35	This work
PM6	3.74	3.65	0.09	This work
D18-Cl	3.83	3.70	0.13	This work
PM6	3.71	3.71	0	Adv. Mater. 34 , 2205926 (2022)
D18-Cl	3.85	3.85	0	Adv. Sci. 9 , 2105347 (2022)
PM6	3.65	3.65	0	Adv. Energy Mater. 11 , 2102000 (2021).
PTzBI-dF	3.62	3.60	0.02	Adv. Funct. Mater. 32 , 2205338 (2022).
D18-Cl	3.92	3.88	0.04	doi.org/10.1002/adfm.202305450

Comment 6: (6) The manuscript should further discuss the physical dynamics, such as the dissociation, recombination, and transport of excitons and charge carriers. There is a lack of characterization to support the claim of "improved charge transport" made in the manuscript.

Response: Thanks for this comment. More characterization and analysis on the exciton and charge dynamics of OSCs are added in the main text as the following “To further investigate the improved performance of OSCs, the exciton dissociation efficiency (P_{diss}) and the charge collection efficiency

(P_{coll}) are calculated and summarized in **Figure S14** and **Table S7** according to previous work^[43, 44]. We found OSCs with INMB-F show negligible change in their P_{diss} but decent improvement in their P_{coll} , suggesting that the ordered molecular packing and shortened π - π stacking preferentially optimize the charge collection process of OSCs. Meanwhile, the dependences of J_{SC} and V_{OC} on light intensity (P_{light}) were further investigated and shown in **Figure S15**. The slopes (α) of all devices are close to unity, suggesting that all OSCs possess relatively weak bimolecular recombination^[26]. The slopes (S) of devices upon INMB-F are all smaller than the control devices, indicating reduced trap-assisted recombination^[34]. Additionally, transient photovoltage (TPV) and transient photocurrent (TPC) further show that all the devices with INMB-F possess longer carrier lifetime and shorter carrier extraction lifetime, validating the effectiveness of more ordered molecular packing and shortened π - π stacking distance for efficient photovoltaic process (**Figure S16**).”

Comment 7: (7) AFM of the active layer morphology should be investigated.

Response: Thanks for this comment. We have further performed AFM and GISAXS to probe the nanoscale morphology of PM6, D18-Cl, PTB7-Th (**Figure S6**) and PM6/L8-BO blend films with or without INMB-F (**Figure S17, S19** and **Table S8**). As shown in **Figure S6**, it is apparent that INMB-F could lead to stronger aggregation and larger domain size for polymers in both their neat and blend films. (**Figure S17, S19** and **Table S8**)

Comment 8: (8) How about the thermal stability of the devices? How about the morphology stability of the active layer?

Response: The thermal and morphological stabilities of PM6/L8-BO devices with or without INMB-F are shown below. (**Picture 6-7** below), and we found the increased structural order and reduced packing distance could stabilize the morphology for OSCs, and leads to enhanced thermal stability of devices.

Picture 6. Performance of PM6/L8-BO and PM6+INMB-F/L8-BO devices upon thermal annealing at 80 °C for different time.

Picture 7. The morphology of PM6/L8-BO and PM6+INMB-F/L8-BO FILMS upon thermal annealing at 80 °C for different time.

Comment 9: (9) The investigation only includes BDT-based polymer donors. It would be beneficial to explore other polymer donors without BDT units as well.

Response: We really appreciate this comment, and hence the effect of INMB-F on P3HT as well PTQ10 were also investigated. However, their absorption spectra are barely affected with the assistance of INMB-F (**Figure S5 (Picture 8 below)**), furthering validating that INMB-F is working on the BDT units. We have rephased our expression in a number of places in the main text that this is applied to BDT-based polymer donors.

Picture 8. (a) Molecular structure of PTQ-10 and P3HT. Absorption spectra of (b) PTQ-10 and (c) P3HT with the addition of INMB-F.

Comment 10: (10) The chemical structures provided in the manuscript should be carefully revised. For example, the structure of D18-Cl is incorrect.

Response: Thanks for this comment. We have corrected the error of structure of D18-Cl in **Figure 1**.

Reviewer #3 (Remarks to the Author):

In this manuscript, a small molecule INMB-F is used to regulate the stacking of polymer donors in films. The π - π stacking distance of polymers is reduced in both pure and blend films. A PCE of 19.4% is realized in PM6/L8-BO based OSCs, which is the highest reported efficiency of binary OSCs. The enhanced structural order of polymer donors by INMB-F also leads to a six-fold enhancement of the operational stability of PM6/L8-BO OSC. The work is novel and interesting, and the manuscript can be accepted for publication after the following comments are addressed.

Comment 1: (1) Strictly say, the system is not binary, since it contains 10% third component INMB-F.

Response: We politely disagree with this comment here. INMB-F is an additive that is to modify structural order, rather than involved in the photovoltaic process. As shown in the **Table** below, the devices only containing PM6 or PM6+INMB-F show negligible difference in their photocurrent, suggesting NMB-F was not involved in the photovoltaic process and only functionalized as the molecular bridge to tune the molecular packing of PM6.

Tab.2. Performance of PM6 with and without INMB-F.

system	treatment	PCE (%)	FF (%)	Jsc (mA cm ⁻²)	Voc (V)
PM6	80 °C TA	0.01±0(0.01)	26.59±0.08(26.69)	0.09±0.01(0.1)	0.409±0.064(0.473)
PM6+INMB-F	80 °C TA	0.01±0(0.01)	27.07±0.83(28.62)	0.05±0.03(0.08)	0.376±0.065(0.518)

Comment 2: (2) How about the miscibility between INMB-F and the three polymers? This should be determined and discussed in the main text.

Response: The miscibility of INMB-F with donors were characterized by contact angle (**Figure S3**), and the corresponding surface free tension (γ) and Flory–Huggins interaction parameter (χ) are summarized in **Table S1**, suggesting that INMB-F is miscible with all polymers.

Comment 3: (3) The miscibility between INMB-F and the acceptors should also be determined and discussed in the main text.

Response: Like discussed above, the miscibility of INMB-F with acceptors was characterized, suggesting that INMB-F is miscible with the acceptors.

Comment 4: (4) How about the charge mobility of polymer films without and with 10% INMB-F?

Response: The hole mobilities (μ_h) of three donors are measured by space-charge-limited current (SCLC) method. As shown in **Figure S10** and **Table S2**, the μ_h of devices are all improved with the addition of INMB-F.

Comment 5: (5) How about the charge mobility of blend films without and with 10% INMB-F?

Response: The charge mobilities of blend films are measured by SCLC methods. More balanced and faster hole (electron) mobilities are obtained (**Figure S13 and Table S6**).

Comment 6: (6) Since the authors claimed that 19.4% is the highest efficiency for binary organic solar cells, the certified efficiency is therefore needed to provide.

Response: Thanks for this comment. We have checked all the certification authority which we could have access to. However, all of them are fully booked within the next 2 months. We hope the reviewer can understand our situation. We no longer claim record PCE in our manuscript.

Comment 7: (7) Can INMB-F influence the morphology, such as fiber width and domain size etc., of blend films? TEM and AFM characterizations of blend films should be provided.

Response: Thanks for this comment. We have further performed AFM and GISAXS measurements to probe the nanoscale morphology of PM6, D18-Cl, PTB7-Th (**Figure S9**) and PM6/L8-BO blend films with or without INMB-F (**Figure S17, S19 and Table S8**). It is apparent that INMB-F could leads to stronger aggregation and larger domain size for polymers in their neat and blend films.

Comment 8: (8) Since the molecular weight of polymers has a significant influence on the device performance, therefore the molecular weight of polymers used should be provided.

Response: Thanks for this comment. The molecular weight of polymers is provided in **Table S1** in supporting information.

Comment 9: (9) How about the nonradiative energy losses without and with INMB-F?

Response: The non-radiative energy losses ($E_{loss}^{non-rad}$) of device with or without INMB-F were calculated through the following equation $E_{loss}^{non-rad} = V_{OC}^{rad} - V_{OC}$, in which V_{OC}^{rad} was carried out by

$$V_{OC}^{rad} = \frac{kT}{q} \ln \left(\frac{J_{SC}}{J_0^{rad}} + 1 \right) \cong \frac{kT}{q} \ln \left(\frac{q \cdot \int_0^{+\infty} EQE(E) \cdot \Phi_{AM1.5G}(E) \cdot dE}{q \cdot \int_0^{+\infty} EQE(E) \cdot \Phi_{BB}(E) \cdot dE} \right),$$

where the k is Boltzmann's constant, T is the temperature of the solar cell ($T = 300$ K is used in this paper), q is elementary charge, $\Phi_{AM1.5G}(E)$ and $\Phi_{BB}(E)$ are the standard solar spectrum under AM1.5G (100 mW/cm²) and black body spectrum at the temperature T of the solar cell, respectively. As shown in the **Table** below, all the devices exhibit similar non-radiative energy loss.

Tab.3. Detailed energy losses of the OPVs based on four OSCs.

OSCs	qV_{oc}^{rad}	E_{loss}^{rad}	$E_{loss}^{non-rad}$	E_{loss}
PM6/C5-16	1.089	0.330	0.258	0.588
PM6+INMB-F/C5-16	1.086	0.338	0.255	0.593
PM6/L8-BO	1.127	0.320	0.247	0.567
PM6+INMB-F/L8-BO	1.124	0.328	0.244	0.572
D18-CI/L8-BO	1.141	0.317	0.230	0.547
D18-CI+INMB-F/L8-BO	1.140	0.314	0.231	0.545
PTB7-Th/C5-16	1.088	0.326	0.412	0.738
PTB7-Th+INMB-F/C5-16	1.089	0.331	0.409	0.740

Comment 10: (10) How about the absorption of INMB-F, can it contribute to the J_{sc} ?

Response: INMB-F only shows absorption ranging from 300 to 500 nm (**Picture 9**), and it could not contribute to J_{sc} in device as the PM6+INMB-F device did not show improved J_{sc} compared to PM6 device (**Tab.2.** in page 14 of response).

Picture 9. Absorption of INMB-F.

REVIEWER COMMENTS

Reviewer #1 (Remarks to the Author):

I appreciate the authors response. I still have some concerns on the revised manuscript.

1. Regarding the novelty of this work, the authors argued that “few work has been endeavored on exploring the effects of volatilizable additives on tuning the molecular packing of the state-of-the-art polymer”. However, have shown INMB-F could not volatilize in the optimal device condition, and the presence of INMB-F in the active layer could also promote the stability of the corresponding devices. Therefore, INMB-F cannot be considered as volatilizable additive in this work. Solid additives used to the molecular packing of the state-of-the-art polymers have been reported by a few groups. Therefore, I still hold the opinion that the novelty of this work is not high enough.
2. I suggested they provided a certification by an authoritative third party because the current density reported in this work is too high to believe. I understand the authors cannot access a certification authority currently. Alternatively, I suggest they send their samples to another group to verify the high performance they realized, which is very easily accessible for them.
3. Although the authors added ^{13}C NMR and ^{19}F NMR spectra of INMB-F in the supporting information, high-resolution mass spectrum is still essential to verify the chemical structure of a molecule.
4. Regarding the packing direction of molecules, the stronger lamellar diffraction peak appeared in the out-of-plane direction suggested a preferred edge-on packing. Herman’s orientation parameter has been widely employed to quantify the lattice plane orientation distribution (Macromolecules 2014, 47, 1403). I suggest they calculate the Herman’s orientation parameter of the 100-diffraction peak to make it clear which orientation is preferred.
5. PM6 and D18-Cl typically show clear fibrils in the AFM images; however, their images in this work are featureless and cannot see much difference after modified by additive. Besides, the AFM images in Figure S17 missed scale bar to show the image size.
6. The authors argued that the presence of INMB-F in the active layer could promote the stability. However, upon heating process, the performance of PM6/C16-16 device with INMB-F increased faster and then decreased faster than the control device. Such variation trend seems not support the improved stability but improved performance with lower operation stability that more sensitive to the temperature.

Reviewer #2 (Remarks to the Author):

The authors have addressed the concerns and comments point by point, I think it's suitable for publication at this stage.

Reviewer #3 (Remarks to the Author):

The manuscript has been carefully revised, all questions raised by the three reviewers have been properly addressed, and now the manuscript is suitable for publication at present version.

Responses to Reviewers' Comments:

Reviewer #1 (Remarks to the Author):

I appreciate the authors response. I still have some concerns on the revised manuscript.

Comment 1: Regarding the novelty of this work, the authors argued that “few work has been endeavored on exploring the effects of volatilizable additives on tuning the molecular packing of the state-of-the-art polymer”. However, have shown INMB-F could not volatilize in the optimal device condition, and the presence of INMB-F in the active layer could also promote the stability of the corresponding devices. Therefore, INMB-F cannot be considered as volatilizable additive in this work. Solid additives used to the molecular packing of the state-of-the-art polymers have been reported by a few groups. Therefore, I still hold the opinion that the novelty of this work is not high enough.

Response: Thanks for this comment. Whether the additive can be defined as a volatilizable additive depends on the volatile temperature and the processing temperature. The volatilizable temperature of INMB-F is 120 °C (not much higher than other solid additive), whilst our processing temperature was 80 °C, we therefore no longer describe INMB-F as a volatilizable additive.

However, we politely disagree the conclusion of lacking novelty of our work. To the best of our knowledge, although some works have observed that solid additives can be used to optimize the domain size or affect the absorption spectrum of polymer donor, very few work has been done to modulate the molecular packing of the state-of-the-art polymers, especially for reducing the π - π stacking distance of a polymer donor film in out-of-plane which is of great important for charge transport in an OSC. Regards that, an additional Table is made below to summarize the recent works on additives optimizing the morphology of the state-of-the-art polymers.

Table 1. Summary of solid additives to modulate the morphology of polymers

Additive	Effect on donor	Characterization method	References
----------	-----------------	----------------------------	------------

DTBF	(1) I_{0-0}/I_{0-1} ratio increases in the absorption spectrum of PM6, (2) PM6 domain size increases	Absorption, AFM	Small 17, 2103497(2021)
FA-C12	(1) PM6 domain size increases	AFM	Nano-Micro Lett. 15, 92, (2023)
TT-Cl	(1) I_{0-0}/I_{0-1} ratio increases in the absorption spectrum of PM6	Absorption	ACS Energy Lett. 8, 3494– 3503 (2023)
DIB	(1) D18-Cl absorption coefficient increases (2) GIWAXS diffraction intensity enhances in D18-Cl/L8-BO blend film	Absorption, GIWAXS, EPS	Adv. Sci. 9, 2200578 (2022)
BBS	(1) I_{0-0}/I_{0-1} ratio increases in the absorption spectrum of PM6	Absorption	Adv. Mater. 34, 2205926 (2022)
INMB-F	(1) Significant reduced π-π stacking distance is observed in a range of BDT-structured polymer donor; (2) The molecular interaction and the corresponding fundamentals are clarified by a range of characterization and simulation methods	Absorption, AFM, GIWAXS, GISAXS, MDS, Raman spectrum	This work

Comment 2: I suggested they provided a certification by an authoritative third party because the current density reported in this work is too high to believe. I understand the authors cannot access a certification authority currently. Alternatively, I suggest they send their samples to another group to verify the high performance they realized, which is very easily accessible for them.

Response: We have been approaching certification authorities since our first round revision, but failed to have access. However, we have been offered a time slot on Aug. 15th, and have

therefore certified our device at National Photovoltaic Product Quality Inspection & Testing Center (China) and obtained a certified PCE of 18.96% (As shown **Picture 1** below and **Supplementary Figure 14**, $FF = 79.50\%$, $J_{SC} = 27.20 \text{ mA cm}^{-2}$, $V_{OC} = 0.877 \text{ V}$).

 220021349274		 中国认可 国际互认 检测 TESTING CNAS L6673	<h1>Test Report</h1>			Test Report No. AGXB123W00327			
Subject unit	Organic Solar Cells-2		
Manufacturer	Wuhan University of Technology		
Commission unit	Tao Wang		
Inspection category	Commissioned inspection		
 Chengdu Institute of Product Quality Inspection Co., Ltd. National Photovoltaic Product Quality Inspection & Testing Center 			

Chengdu Institute of Product Quality Inspection Co., Ltd.
National Photovoltaic Product Quality Inspection & Testing Center
TEST REPORT

Test Report No. AGXB123W00327

Page 1 of 2

Product Name	Organic solar cells-2	Trade Mark	/
Manufacture Date	14/08/2023	Model /Type	0.0404cm ²
Sample No.	AGXB123W00327	Sample Grade	/
Sample Quantity	One piece	Sample State	/
Delivery Date	15/08/2023	Sample Delivered personnel	Tao Wang
Commission unit	Tao Wang	Manufacturer	Wuhan University of Technology
Commission unit address	School of Materials Science and Engineering, Wuhan University of Technology, No. 122 Luoshi Road, Wuhan 430070, P. R. China	Manufacturer Address	No. 122 Luoshi Road, Wuhan 430070, P. R. China
Commission unit Zip code	430070	Manufacturer Zip code	430070
Commission unit Tel.	18507131726	Manufacturer Tel.	18507131726
Center Address	No. 355, 2 nd Tengfei Road, Southwest Airport Economic Development Zone, Chengdu, Sichuan, P. R. China.	Measurement Date	15/08/2023
Methods	IEC 60904-1:2020 Photovoltaic devices-Part 1: Measurement of Photovoltaic Current-Voltage Characteristics.		
Test conclusion	This column blank.		
Remarks	The mask area is provided by the Commission unit: 0.0404cm ² .		
Approved by	成皓楠	Reviewed by	许维
		Measured by	游宗英

Special chapters for test reports
Issue Date: 15/08/2023

Chengdu Institute of Product Quality Inspection Co., Ltd.
National Photovoltaic Product Quality Inspection & Testing Center
TEST REPORT

Test Report No. AGXB123W00327

Page 2 of 2

Test Results:

No.	Test item(s)	Unit	Results
1	Current-voltage characteristics measurement	---	---
1.1	Open-circuit voltage, V_{oc}	V	0.877
1.2	Short-circuit current, I_{sc}	mA	1.099
1.3	Maximum-power, P_{max}	mW	0.766
1.4	Maximum-power voltage, V_{p-max}	V	0.749
1.5	Maximum-power current, I_{p-max}	mA	1.023
1.6	Fill factor, FF	%	79.50
1.7	Conversion efficiency, η	%	18.96

Current-voltage characteristics

Remark: Sample was tested under the irradiation with a steady-state class calibrated AAA solar simulator (AM1.5-G 1000.0 W/m² based on mono-Si reference cell) at 25 ± 1 °C. Designated area defined by thin metal aperture mask.

Blank

Picture 1. Certification by the National Photovoltaic Product Quality Inspection & Testing Center (China). The device was measured with a mask of 4.04 mm² and gave a PCE of 18.96%.

Comment 3: Although the authors added ^{13}C NMR and ^{19}F NMR spectra of INMB-F in the supporting information, high-resolution mass spectrum is still essential to verify the chemical structure of a molecule.

Response: Thanks for this comment. The high-resolution mass spectrum has been added (see picture 2 below and Supplementary).

Picture 2. The high-resolution mass spectrum of INMB-F.

Comment 4: Regarding the packing direction of molecules, the stronger lamellar diffraction peak appeared in the out-of-plane direction suggested a preferred edge-on packing. Herman's orientation parameter has been widely employed to quantify the lattice plane orientation distribution (Macromolecules 2014, 47, 1403). I suggest they calculate the Herman's orientation parameter of the 100-diffraction peak to make it clear which orientation is preferred.

Response: We appreciate this suggestion, and have calculated the Herman's orientation parameter (S) of the (100) diffraction peak of our PM6 and PM6+10% INMB-F films via the above method. As shown in picture 3 below (Supplementary Fig. 10), the calculated S of PM6 and PM6+10% INMB-F films are 0.47 and 0.60 respectively, suggesting INMB-F has affect the crystallization of PM6 profoundly and even leads to a more preferred edge-on orientation.

Picture 3. Quasi-pole image and plots of the (100) lattice plane of PM6 and PM6+INMB-F films.

Comment 5: PM6 and D18-Cl typically show clear fibrils in the AFM images; however, their images in this work are featureless and cannot see much difference after modified by additive. Besides, the AFM images in Figure S17 missed scale bar to show the image size.

Response: Thanks for the comment. In fact, the nanoscale morphology of polymer donors can be highly affected by their molecular weight and dispersity index, and the absence of fibril domains of PM6 and D18-Cl have also been reported in other literatures as shown in **Picture 4** below. The missed scale bar has been added in **Supplementary Fig. 19** (Supplementary Fig. 17 is ordered into Supplementary Fig. 19 after revision).

Picture 4. The AFM images of PM6 and D18-Cl films reported in literature (Adv. Energy Mater. 12, 2200165 (2022) and (b) Sol. RRL 7, 2201084 (2022)).

Comment 6: The authors argued that the presence of INMB-F in the active layer could promote

the stability. However, upon heating process, the performance of PM6/C5-16 device with INMB-F increased faster and then decreased faster than the control device. Such variation trend seems not support the improved stability but improved performance with lower operation stability that more sensitive to the temperature.

Response: Thanks for this comment. We agree with the Reviewer that the performance of our PM6/C15-16 increases faster within the temperature below 80 °C and decreases faster when temperature over 120 °C, which further indicate that the exist of INMB-F can be used to stabilize the morphology of active layer, as the power conversion efficiency of devices with or without INMB-F dropped to the same level upon 150 °C when all the INMB-F molecules are evaporated.

On the other hand, during the operation of OSCs, their temperature can hardly reach over 100 °C, which means it will be difficult for our INMB-F to volatilize and we believe that's the reason why INMB-F can lead to enhanced operational stability and thermal stability upon 80 °C (see picture 5b below).

Picture 5. Efficiency variation of (a) PM6/C5-16 OSCs with and without INMB-F under different thermal annealing temperature and (b) PM6/L8-BO OSCs upon thermal annealing at 80 °C for 350 mins.

Reviewer #2 (Remarks to the Author):

The authors have addressed the concerns and comments point by point, I think it's suitable for publication at this stage.

Response: We appreciate to the reviewer for supporting this research to be published in Nature Communications.

Reviewer #3 (Remarks to the Author):

The manuscript has been carefully revised, all questions raised by the three reviewers have been properly addressed, and now the manuscript is suitable for publication at present version.

Response: We appreciate to the reviewer for supporting this research to be published in Nature Communications.

REVIEWERS' COMMENTS

Reviewer #1 (Remarks to the Author):

The authors have addressed all the questions. This work can be accepted without any change.